# What Is the Role of Industry-Based Intermediary Organisations in Supporting Workplace Mental Health in Australia? A Scoping Review

**DOI:** 10.3390/ijerph22070974

**Published:** 2025-06-20

**Authors:** Kristy Burns, Louise A. Ellis, Abilio De Almeida Neto, Carla Vanessa Alves Lopes, Janaki Amin

**Affiliations:** 1Department of Health Systems and Populations, Macquarie University, North Ryde, NSW 2113, Australia; 2SafeWork New South Wales, Gosford, NSW 2250, Australia

**Keywords:** psychological health, workplace mental health, intermediaries, scoping review

## Abstract

Despite increasing interest in workplace mental health, limited attention has been paid to the role of industry-based intermediary organisations in delivering mental health support. This scoping review addresses this gap by examining the mental health-related activities of industry intermediaries in Australia. A systematic search of the peer-reviewed and grey literature from 2010 to 2023, supported by expert consultation and conducted in accordance with the PRISMA-ScR guidelines, identified 35 relevant records. Interventions were categorised using the WHO Guidelines on Mental Health at Work and evaluation activities coded according to the Conceptual Framework for Implementation Research. Organisational-level interventions were the most common (54%), followed by individual psychosocial support for distressed workers (40%). Mental health-specific intermediaries offered more WHO-recommended interventions and were more likely to evaluate their programs. Although evaluations suggested improvements in mental health literacy and high program acceptability, evidence of impact on worker health and organisational outcomes was limited. The findings suggest that intermediary organisations—including unions, business associations, and professional bodies—are well positioned to support tailored mental health strategies. However, the current lack of robust evaluations limits the understanding of their effectiveness. Future efforts should prioritise industry–research collaborations to strengthen the evidence base and inform sustainable investment in workplace mental health initiatives.

## 1. Introduction

Promoting mentally healthy workplaces has been identified as one of seven priority actions to improve population mental health [1]. Over the past two decades, research has focused on identifying workplace risk factors and the subsequent design of interventions to improve employee mental health. While there has been a flurry of activity to support worker mental health in workplaces around the globe, there is concern that much of this activity lacks an evidence base [2]. With the aim of increasing evidence-based interventions, the World Health Organization released its first guidelines on mental health at work [3] in 2022.

Current approaches to improving workplace mental health typically occur at two levels: micro-level initiatives implemented within individual businesses and macro-level interventions developed and disseminated at the national or state level. While each approach has value, micro-level initiatives often require substantial internal resources and capacity, limiting their feasibility for small- and medium-sized enterprises (SMEs). Conversely, macro-level initiatives may struggle to address the diversity of industry contexts, limiting their relevance and uptake.

To encourage further action on workplace mental health, an opportunity exists between these approaches at the meso-level, where sector-wide, industry-specific strategies can be supported and scaled through intermediary organisations such as business associations, unions, and professional peak bodies. These intermediaries are well placed to tailor, translate, and disseminate interventions in ways that are contextually appropriate, particularly for SMEs. Their established role in advancing occupational health and safety is well documented, including as trusted conduits between regulators and workplaces [4,5,6]. This role is conceptually grounded in the Regulator–Intermediary–Target (RIT) Model [7], which describes how regulators can engage intermediary actors to influence target groups (in this case, workplaces) through knowledge translation, capacity building, and contextual adaptation. Given the known variation in workforce mental health profiles across industries [8] and that working conditions are known to impact mental health [9], the potential of intermediaries to support industry-specific mental health strategies warrants focused examination. While these groups have increasingly engaged in mental health promotion, the scope and effectiveness of their activities have not yet been systematically synthesised within the academic literature. The limited volume of peer-reviewed studies, coupled with the heterogeneity of available sources—ranging from websites to internal and consultant-authored reports—renders a traditional systematic review or meta-analysis inappropriate at this stage. In contrast, a scoping review is particularly suited to mapping emerging fields where the evidence base is diffuse, predominantly located in the grey literature, and not yet mature enough to support quality appraisal or quantitative synthesis. As such, this review represents an important foundational step in understanding the role of industry-based intermediaries in workplace mental health. This review was conducted in accordance with the PRISMA-ScR guidelines [10] to systematically identify, chart, and categorise mental health initiatives undertaken by Australian industry intermediaries.

This review aims to explore the role of industry intermediary organisations in supporting workplace mental health in Australia. In particular, we seek to understand (i) what types of intermediary organisations provide mental health support to workplaces, (ii) what types of programs are offered to workplaces—mapped to the WHO Guidelines on Mental Health at Work—and (iii) what evidence exists for their effectiveness.

For this scoping review, industry intermediary organisations are defined as external non-profit groups who engage with and support workplaces on a regular basis. Typically, intermediaries represent and support member companies or individuals within a particular industry sector (e.g., the Housing Industry Association) or profession (e.g., the Australian Veterinary Association). Consultants and commercial entities delivering mental health/wellness programs for profit are not in scope for this review.

This review focuses on Australia due to the recognised role that intermediary organisations play in the country’s occupational health and safety landscape [11], supported by well-established structures such as industry-funded bodies and unions. Additionally, this decision was influenced by practical constraints in identifying the grey literature internationally, as Google’s location-based algorithms limited access to non-Australian content. This geographical focus enables a more comprehensive and contextualised synthesis of mental health activities within a national system where intermediaries have demonstrable influence.

## 2. Materials and Methods

The methods for this scoping review were guided by the framework created by Arskey and O’Malley [12], and the results are reported according to the Preferred Reporting Items for Systematic Reviews and Meta-Analyses checklist for scoping reviews (PRISMA-ScR) [10]. A protocol outlining key review parameters was registered on the Open Science Framework in August 2023 (Available online: https://osf.io/kbzfd/ (accessed on 10 January 2025)).

### 2.1. Information Sources and Search Strategy

The search strategy was developed in consultation with an academic research librarian and centred on the keywords “workplace”, “mental health”, and “intermediary”, including their relevant synonyms. The strategy incorporated the following components:Bibliographic databases: A search of Medline, Embase, Scopus, and Business source premier, from 1 January 2010 to 1 December 2023, was undertaken, along with one on Google Scholar (first 200 citations). The search strategy for Medline was as follows (adapted for other databases):exp workplace/.exp mental health/.1 AND 2.(advocacy or intermediar * or union or trade or profession * or mediator * or facilitat * or arbitrat *). tw.3 AND 4.Limit 5 to (yr = “2010-current”).A Google search (first 200 results) was conducted to identify the relevant organi-sations and websites publishing content on the subject area. Next, each of the rel-evant websites’ homepage were manually searched for potentially relevant doc-uments (e.g., web pages, reports). Within this step, each website and the date on which each search was conducted were documented.A targeted website search was conducted, including industry intermediaries identified in the Google search (ii) above, along with a review of policy and guideline documents from intermediary websites and government bodies.Manual searches on the reference lists of previously identified documents were conducted.Relevant experts were consulted.

Whilst the protocol outlined searching opengrey.eu, this service was discontinued in 2020. The review protocol also described an international review; however after commencing the search, it was apparent that Google’s location-based results algorithm meant limited international results were forthcoming. To support a comprehensive review, the search was limited to Australia.

### 2.2. Eligibility Criteria and Intermediary Classification

Records were included if they were reporting on intervention programs designed to improve workforce mental health (including depression, anxiety, trauma, wellness, or reducing suicide), focused on a specific industry or profession, undertaken in Australia, and published in English. Descriptive data about the industries and interventions were extracted.

Records were excluded if the intervention programs were delivered by commercial (for-profit) entities, were broad interventions for any industry or workplace, did not include a significant industry intermediary role, were focused on a single workplace, or were focused solely on return-to-work, with no prevention component.

Intermediaries were classified according to the International Classification of Non-profit and Third Sector Organizations (ICNPO) 2017 [13]. Non-profit status criteria were as per the ICNPO: i. private (not controlled by government); ii. main purpose is to serve a social or public purpose rather than to maximise and distribute returns on invested capital; and iii. engagement in them is voluntary (performed by free will and without coercion). Figure 1 shows the in-scope determination and classification sequence for intermediaries.

### 2.3. Selection

Titles and abstracts from the literature search were managed using Covidence and screened independently by two reviewers (K.B. and C.L.) according to the eligibility criteria. The first five records identified via each of the above sources were screened by both reviewers to ensure agreement, with discrepancies discussed until consensus was reached. Two reviewers independently screened 5% of full-text review records, with conflicts resolved though discussion and clarification, followed by full-text screening conducted independently by the same reviewers for the remaining records. The PRISMA flow chart in Figure 2 shows reasons for exclusion at each stage of this review.

### 2.4. Data Extraction and Analysis

Data were extracted and put into Excel spreadsheets and included the following: citation, record type, industry/profession, intermediary type, WHO guideline program type, and evaluation outcome type. Evaluation type was coded according to Proctor et al.’s [14] Conceptual Framework for Implementation Outcomes. In accordance with Proctor et al.’s [14] model, outcomes were distinguished at three levels: implementation; service; and client outcomes. For the application of these outcomes to workplace research, we conceptualised these constructs as follows: implementation outcomes (e.g., outcomes of the implementation process); organisational outcomes (e.g., organisational change, productivity impacts); and individual outcomes (e.g., impact on mental health, mental health literacy, other health outcomes, and user satisfaction).

Program type was coded based on the WHO Guidelines on Mental Health at Work [3] recommendations. When the WHO-recommended intervention type had a specific target of humanitarian and emergency workers, the category was collapsed into the general worker category of this type. Returning to work programs were not included as they were outside the scope of this review. Final program type categorisation was as follows: (i) Organisational: universal interventions. (ii) Organisational: interventions for workers with mental health conditions. (iii) Manager training for mental health knowledge and behaviour. (iv) Worker training in mental health literacy. v. Individual: universally delivered psychosocial interventions. (vi) Individual: universally delivered physical activity programs. (vii) Individual: psychosocial interventions for workers with emotional distress. (viii) Individual: physical programs for workers with emotional distress.

## 3. Results

Of the 35 records included in this review, 5 were peer-reviewed journal articles, 4 were published reports authored by external academic consultants, 7 were published reports authored by internal program staff, and 19 were websites.

### 3.1. Classification of Organisations

Organisations offering mental health support varied in their classification according to the ICNPO [13]. Improving mental health was the primary purpose of 15 industry intermediaries (classified as 03400: other health services). Three intermediaries were labour unions (11300), eleven were business associations (11100), and five were professional associations (11200). One intermediary was a workers entitlement scheme (classified as “other”).

### 3.2. Description of Programs

The interventions offered by intermediaries varied greatly. Whilst many offered interventions aimed at changes at the organisational level, the program content varied from single policy recommendations to comprehensive program development support. Individual psychosocial programs for workers with emotional distress were also common, and almost all consisted of free crisis counselling for workers. Table 1 summarises the 35 programs offered by industry intermediaries included in this review, including the type of intermediary and intervention type categorised according to the WHO recommendation categories.

### 3.3. Intervention Types

Figure 3 (below) charts the proportion of intermediaries offering each of the intervention types according to the WHO Guidelines on Mental Health at Work. The most common type of intervention offered by intermediaries was organisational: universal, with 54.3% (*n* = 19) of intermediaries offering at least one intervention of this type. Worker training (*n* = 16, 45.7%) and manager training (*n* = 13, 37.1%) were also commonly offered. Individual psychosocial interventions for workers experiencing emotional distress were offered by 40% of organisations (*n* = 14).

In addition to these, just under half of all intermediaries (*n* = 15, 45.7%) offered an intervention that did not directly relate to a WHO recommendation. In most cases these were links to external information and crisis support services.

#### Number of Interventions Offered by Intermediary Classification

The overall mean number of WHO guideline interventions offered by intermediaries was 2.1. A total of 11 of the 35 intermediaries offered just one intervention type (i.e., only worker training: 31.4%). Mental health-specific organisations offered a mean of 2.5 intervention types, followed by unions with a mean of 2.3. Professional associations offered a mean of 1.8 intervention types, while business associations offered a mean of 1.3. The combined mean for all non-mental health-specific organisations (business, professional, and unions combined) was 1.7 WHO interventions.

### 3.4. Evaluation Activities

Figure 4 shows the range of evaluation activities being conducted by intermediaries on their mental health interventions. Note that some intermediaries undertook multi-component evaluations which included more than one type of evaluation. Of the 35 intermediaries, 14 had documented some evaluation of their mental health programs (40%). Over half (*n* = 21 of 35, 60%) had no documented evaluation. Implementation (process) outcomes were most commonly measured (in 13 evaluations), followed by individual-level outcomes. Organisational-level changes were evaluated the least, with two evaluations measuring policy change and three measuring other organisational outcomes. Two evaluations included “other” components: an economic evaluation and a retrospective mortality study.

#### Proportion of Intermediaries with Published Evaluation of Their Mental Health Program, by Intermediary Classification

Of the 15 mental health-specific intermediaries, 8 reported on some type of evaluation of their programs (53.3%). Four of the eleven business associations (36.4%) and one of the three unions (33.3%) reported on an evaluation. None of the five professional associations reported evaluations.

The combined proportion of all non-mental health-specific organisations (i.e., unions, professional, and business associations) undertaking some type of evaluation was 30%.

### 3.5. Evaluation Outcomes

#### 3.5.1. Implementation Outcomes

Of the 14 programs that were evaluated, 13 reported on reach (penetration) outcomes, describing program involvement in terms of the number of participants or workplaces. User perspective (acceptability) was measured by either user satisfaction items or a rating of the likelihood of recommending the program to others. Eight evaluations included a measure of user perspective [15,20,33,42,43,44,48,49]. For each, user satisfaction was high. Of the five evaluations that measured the likelihood of the user recommending the program to others, over 80% responded yes in each of the evaluations [20,42,44,48,49].

#### 3.5.2. Organisational Outcomes

Two evaluations reported on policy changes following program implementation. In an evaluation of the impact of the Mates in Mining program, Tynan [48] reported positive changes to the worker perception of mental health policies following the introduction of the program. Using a policy audit methodology, Schroder [15] reported on improvements to workplace policy following the program; however quantitative data was not described.

Three programs described other organisational changes as a result of program implementation. One survey evaluation reported on improvements in employee perception of workplace domains such as leadership and culture post-implementation, although statistical analyses were not conducted [33]. Another analysed changes to the worker perception of organisational commitment to mental health, including the availability of flexible work arrangements and mental health training for employees and managers, and found a significant positive change in the perception of workplace commitment to mental health in each domain except flexible work arrangements [48]. One qualitative study described an increase in trauma prevention actions in workplaces post-intervention, including the implementation of reflective practice sessions and wellbeing plans as part of onboarding and performance reviews [15].

#### 3.5.3. Individual Outcomes

Four evaluations measured the impact on worker knowledge using a pre/post-survey methodology [33,44,48,49], and each demonstrated an increase in worker knowledge/mental health literacy. In two qualitative analyses of participant interviews, both found an increase in capability and motivation to help others following participation in the program [50,51].

Manager/supervisor knowledge and confidence in supporting workers with mental health problems were found to increase following program involvement in two studies. A pre/post-survey analysis [48] found a statistically significant increase in manager mental health literacy following program completion, while a qualitative analysis of interviews with team leaders found an increase in knowledge around mental health symptoms and greater confidence to discuss metal health with team members following training [33].

Three of the five evaluations analysing changes in mental health outcomes used the validated Kessler psychological distress scale [52] in pre/post-single-arm studies, with mixed findings. One study found that the post-intervention distress scores of workers were significantly lower than baseline and that this was maintained at the 6 month follow-up [42]. One evaluation showed a small (one-point) increase in the distress score in for workers following participation; however the small sample size precluded statistical analysis [33], and it is unlikely that a one-point change is clinically meaningful [53]. Finally, Sayers [47] found no change in psychological distress in pre/post-evaluations; however, again, statistical analyses were not conducted on this data.

Two evaluations measured changes to mental health outcomes using a single “emotional wellbeing” item with a 5-point scale (1 = very poor; 5 = very good), and both found a significant improvement in wellbeing rating from pre- to post-intervention [44,49]. One of these included a follow up analysis and found no difference in wellbeing from pre-intervention to 3-6-month follow-up [44].

One program evaluation measured changes to other health outcomes and showed an increase in the number of workers who reported exercising at least three times a week and eating a healthy diet in a pre/post-evaluation [33].

#### 3.5.4. Other Evaluation Types

For one program, a Social Return on Investment (SROI) economic evaluation was conducted, with the findings estimating an AUD 2.44 “social value return” for every dollar invested in the program [35]. Unfortunately, this program included multiple components, including short-term financial support, and the evaluation was unable to attribute returns to individual components. In a retrospective mortality study on suicide rates, age-standardised suicide rates were shown to have declined significantly more amongst construction workers compared with other employed men between 2001 and 2019, and the authors conclude that sector-specific suicide prevention efforts (including MATES) may have contributed to this decline [46].

## 4. Discussion

Industry intermediaries in Australia are delivering a range of programs aimed at improving the mental health of their respective workforces. More than half are offering organisational-level intervention support, aimed at structural change to achieve mentally healthy workplaces, although individually focused interventions are also common. While the majority of programs did not have documented evaluation evidence, the evidence from those that did conduct evaluations suggests they are well received and can have a positive impact on worker and manager mental health knowledge and skills. There are preliminary indications of beneficial effects on employee wellbeing, albeit with limited empirical evidence. Considering the demonstrated efficacy of industry intermediaries in facilitating the uptake of workplace health and safety practices in diverse contexts [54,55], the emergence of a concerted emphasis on mental health initiatives by numerous industry intermediaries in Australia is an encouraging development.

While there is a growing recognition of the value of incorporating both organisational-level and individual-level interventions [56,57], and meta-analytic findings lend considerable support for multi-intervention approaches [58], our research suggests that intermediaries are offering a limited array of intervention types, with 31% offering only a single intervention, i.e., worker training. Particularly noteworthy is the finding that mental health-specific intermediaries were more likely to offer multi-intervention programs with more intervention types than business or professional associations and unions, presumably due to their primary focus on mental health. Nevertheless, the Thrive [32] program implemented by the Australian Veterinary Association exemplifies a professional association that offers a comprehensive range of intervention types, both at the organisational and individual levels, suggesting that the adoption of comprehensive approaches is feasible for unions, business, and professional associations.

More promising is the finding that the predominant program type offered by industry intermediaries consists of universal organisational interventions. Universal organisational interventions (Recommendation 1: WHO Guidelines on Mental Health at Work) are organisational interventions delivered to an entire workforce to address psychosocial risk factors and reduce emotional distress, such as job redesign, improving role clarity, and increasing worker control [3]. Despite various workplace intervention studies demonstrating the effectiveness of organisational-level approaches centred on restructuring the work environment [9,59], and formal recommendations to prioritise this approach over individual illness-oriented strategies [60,61], interventions that target the individual remain the most common [56]. Our finding that more than half of Australian industry intermediaries examined in 2023 are providing mental health initiatives aligned with organisational-level change is gratifying to observe. In fact, given that knowledge of population-level contributing factors is important for mitigating poor mental health [62], and that a key barrier to uptake in individual businesses is knowledge gaps [54], the industry-specific knowledge of industry intermediaries puts them in a unique position to advocate for and support organisational-level wellbeing strategies.

A significant gap in evaluation activity was identified, with few intermediaries engaging in evaluations extending beyond measures of reach and user satisfaction and fewer incorporating a control group. Without rigorous evaluation, stakeholders cannot ascertain whether or not investments in these programs are yielding meaningful health and/or workplace returns, hindering informed decision-making and potentially squandering valuable resources while allowing mistakes to be repeated. The best evaluation evidence comes from industry–research partnerships, namely the MATES suite of programs [45] and ifarmwell [42]. These collaborations have yielded multiple peer-reviewed publications, offering a platform for ongoing program enhancement and knowledge sharing, and provide a model for future endeavours.

Clearly, certain industry groups in Australia are energetically addressing mental health, while others show limited action. For those in the latter category, especially when their workforce profile indicates high levels of worker distress [8], there is a lot to learn from more successful industry intermediaries. To facilitate this information exchange, prioritising the rigorous evaluation of current industry-led mental health initiatives is crucial, even as we acknowledge the inherent challenges involved. Impact evaluations to determine whether a causal relationship exists between programs and individual health outcomes are notoriously difficult and resource-intensive to undertake for work health and safety programs [54]. The industry–workplace–researcher partnership model can contribute significant value here, as industry knowledge combines with research know-how to yield more efficient outcomes. Importantly, successful collaboration requires a “transactional” approach, where stakeholders both contribute to and gain equally from the partnership [14]. Funders should look to include mandatory evaluations, including the publication of findings, in funding agreements, along with the financial resources to support this.

Also interesting to note is that despite the prevalence of industry intermediaries offering organisational-level interventions, changes at the organisational level are rarely being evaluated, which is consistent with workplace mental health research internationally [57]. Lemke et al. [57] have argued that intervention research that evaluates changes in work organisation and policy change should be prioritised as they are more likely to generate sustainable change. Given the known challenges to health impact evaluations in work health and safety, it seems an opportunity exists to increase the measurement of organisational-level changes resulting from industry intermediaries’ actions in mental health, including changes to policy and the work environment. Further measuring the impact of mental health programs on productivity, including conducting economic analyses, would be particularly beneficial for engaging business who are otherwise delaying action on workplace mental health.

The inclusion of grey literature was both a strength and a limitation of this research. Considering the scarcity of peer-reviewed evidence on the topic, the grey literature emerged as an essential source of information, enabling a comprehensive review of the mental health support activities provided by industry groups. It also allowed for the inclusion of more contemporary material, which has been recognised as especially important in organisational research [63]. In addition, the inclusion of grey literature provides an opportunity to include evaluations with null or negative results that might not otherwise be published, thus reducing publication bias [64]. Conversely, methodological rigour is not guaranteed with grey literature. However, given that the general purpose of conducting a scoping review is to identify and map the available evidence [12], excluding the grey literature in this case would be at odds with this aim.

The application of the WHO Guidelines on Mental Health at Work [3] as a framework to categorise the interventions being offered by intermediaries is a strength of this research, as it provides a standardised evidence-informed structure and allows for meaningful assessment against recommended actions, highlighting gaps for further action. We did not undertake a quality assessment of individual evaluation studies, as is typical with scoping reviews. We also note that we did not include evaluations that are not available in the public domain. Furthermore, although return-to-work interventions constitute a category within the WHO guidelines, they were excluded from this review in alignment with our focus on upstream mental health strategies aimed at primary and secondary prevention. Nonetheless, intermediaries may play a valuable role in facilitating return-to-work processes, and further investigation into this potential contribution is warranted.

Lastly, this research was conducted within the context of Australian organisations as a result of international web search limitations for grey literature. While this enabled a detailed understanding of the current status of industry-led mental health approaches within the Australian landscape, the applicability of these findings to other countries may be limited. Future research could aim to replicate this study in different national contexts to compare the findings, thereby enriching the global understanding of the issue.

## 5. Conclusions

This scoping review systematically mapped the mental health-related activities of industry intermediary organisations in Australia, addressing a significant gap in understanding how these entities support workplace mental health initiatives. This review identified that intermediary organisations are actively engaged in delivering workplace mental health interventions, with organisational-level approaches being the most prevalent, with individually focused interventions also being common.

However, a key limitation identified across the field is the lack of robust evaluation practices, with evidence of measurable impact on worker health outcomes and organisational effectiveness remaining limited despite programs showing promise in improving mental health literacy and achieving high acceptability rates. The scoping review approach proved particularly valuable for synthesising this emerging field where evidence is predominantly located in the grey literature. Moving forward, strengthening industry–research collaborations will be essential to develop rigorous evaluation frameworks that can demonstrate the effectiveness of intermediary-led mental health initiatives and support sustainable investment in workplace mental health strategies.

## Figures and Tables

**Figure 1 ijerph-22-00974-f001:**
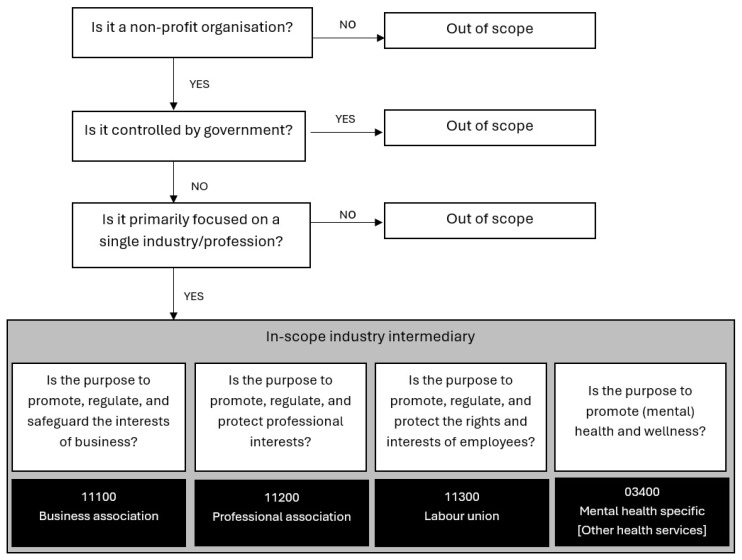
Sequence of steps for determining in-scope organisations and ICNPO classification.

**Figure 2 ijerph-22-00974-f002:**
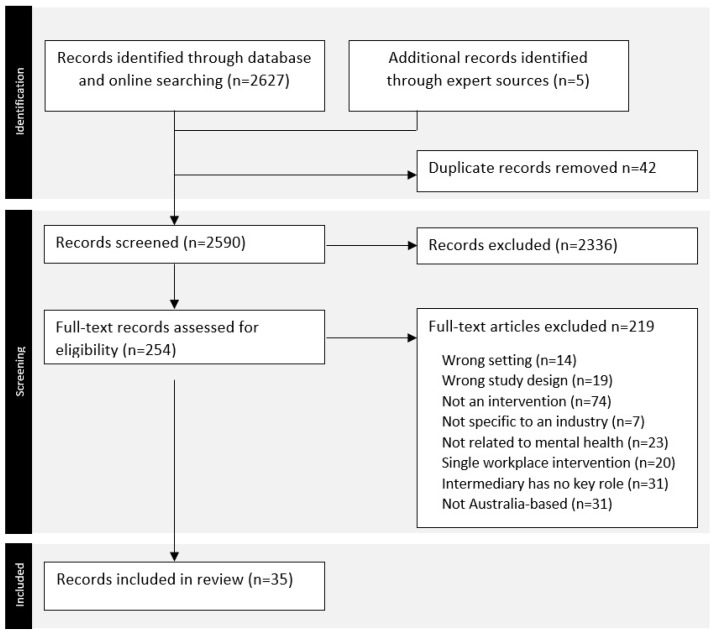
PRISMA flow chart.

**Figure 3 ijerph-22-00974-f003:**
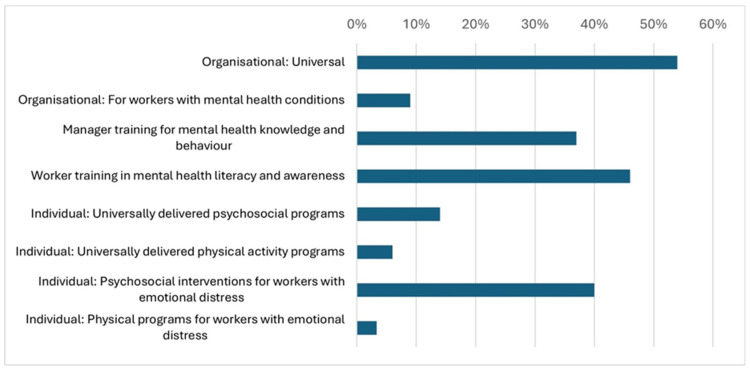
WHO-recommended interventions: proportion of intermediaries offering each activity.

**Figure 4 ijerph-22-00974-f004:**
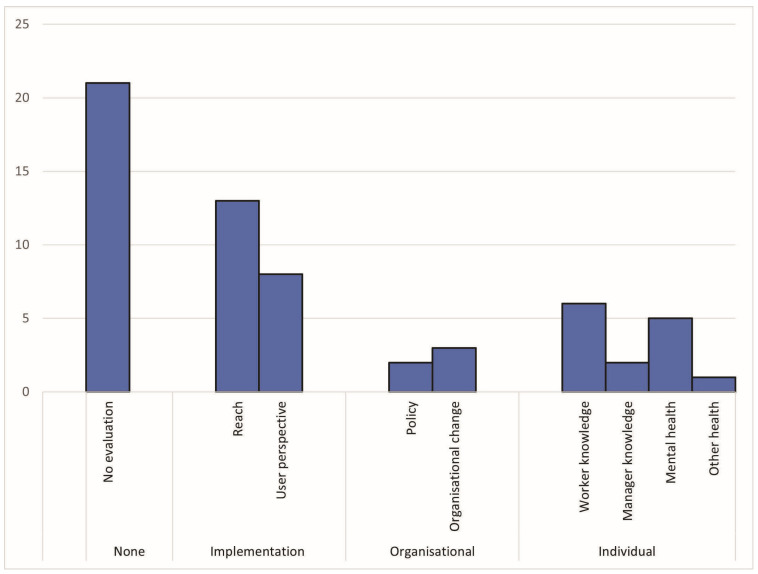
Number of intermediaries conducting evaluation activities by evaluation type.

**Table 1 ijerph-22-00974-t001:** Summary of workplace mental health programs offered by intermediaries.

Program	Author, Year	Intermediary Type ^a^	Intervention Type (s): WHO Recommended ^b^	Evaluated Y/N
Vicarious trauma prevention and awareness toolkit Available online: http://vtpat.org.au/ (accessed on 18 January 2024)	Schroder [15] 2022	U	OU, OMH	Y
Collaboration delivers wellbeing: Australian Manufacturing Workers Union Collaboration Available online: https://www.amwu.org.au/workwell_resources (accessed on 17 January 2024).	Australian Manufacturing Workers Union [16]	U	OU, MT, WT, IPSD	N
Stand TALR. WA Prison Officers Union Available online: https://standtalr.org/ (accessed on 15 January 2024).	Western Australian Prison Officers’ Union [17]	U	WT	N
Fine Tuning Automotive Mental Health Available online: https://www.vaccsdc.com.au/projects/fine-tuning-automotive-mental-health-project/ (accessed on 16 January 2024).	Victorian Automotive Chamber of Commerce [18]	BA	OU	Y
Water Industry Mental Health Framework Available online: https://www.wsaa.asn.au/ (accessed on 24 January 2024).	Water Services Association of Australia [19]	BA	OU	N
Stay Afloat Available online: https://www.stayafloat.com.au (accessed on 15 January 2024).	Williams and Fildes [20] 2021	BA	OU, IPSD, O	Y
Building Mentally Healthy Environments: HIA Workplace toolkit Available online: https://hia.com.au/hia-community/what-we-do/mental-health-program (accessed on 22 January 2024).	Housing Industry Association [21]	BA	MT, WT, O	N
AFL Industry Mental Health & Wellbeing Strategy Available online: https://www.afl.com.au/mental-health-wellbeing/strategy (accessed on 15 January 2024).	Australian Football League [22]	BA	OU, O	N
Consult Australia: Striving for mentally healthy workplaces Available online: https://www.consultaustralia.com.au/home/business-services/mental-health-knowledge-hub (accessed on 21 January 2024).	Consult Australia [23]	BA	OU, O	N
Resources and energy industry: Workforce mental health framework Available online: https://www.areea.com.au/framework (accessed on 15 January 2024).	Australian Resources and Energy Employer Association [24], 2021	BA	OU, OMH, MT, WT, O	N
Grain Producers Australia: Farmer Mates Mental Health Available online: https://www.grainproducers.com.au/farmer-mates-mental-health (accessed on 26 January 2024).	Grain Producers Australia [25]	BA	O	N
Australian Retailers Association: Mental wellbeing Available online: https://www.retail.org.au/mental-wellbeing (accessed on 21 January 2024).	Australian Hotels Association (SA) [26]	BA	O	N
Australian Hotels Association South Australia: Check Inn Available online: https://www.ahasa.com.au/latest-news/mental-health-and-wellbeing-program (accessed on 30 January 2024).	Australian Hotels Association (SA) [26]	BA	IUPS	N
Civil Contractors Federation: Positive Plans Positive Futures Available online: https://www.ccfvic.com.au/positive-plans-positive-futures (accessed on 20 January 2024).	Civil Contractors Federation Victoria [27]	BA	OU	Y
Chartered Accountants Wellbeing Available online: https://www.charteredaccountantsanz.com/member-services/mentoring-and-support/ca-wellbeing (accessed on 15 January 2024).	Chartered Accountants Aus/NZ [28] 2024	PA	IPSD, O	N
Association of Consulting Architects: Wellbeing Available online: https://aca.org.au/knowledge-hub/other-key-issues/mental-wellbeing (accessed on 17 January 2024).	Association of Consulting Architects Australia [29]	PA	OU, MT, ISPD, O	N
Australian Institute of Architects: Mental Health Practice Notes Available online: https://acumen.architecture.com.au/practice/human-resources/mental-health-in-the-profession (accessed on 20 January 2024).	Australian Institue of Architects [30]	PA	O	N
Australian & International Pilots Association: Pilot Welfare program Available online: https://aipa.org.au/welfare (accessed on 21 January 2024).	Australian and International Pilots Association [31]	PA	IPSD, O	N
Thrive: Veterinary industry wellness program Available online: https://www.ava.com.au/Thrive/ (accessed on 18 January 2024).	Australian Veterinary Association [32], 2022	PA	OU, MT, WT, IPSD, O	N
Wellbeing On Call: Thriving Contact Centres Previously at: https://wellbeingoncall.superfriend.com.au (accessed on 18 January 2024).	Superfriend [33], 2020	MHS	OU, OMH, MT, WT, IUPS, IUPA	Y
Entertainment Assist Available online: https://www.entertainmentassist.org.au (accessed on 16 January 2024).	Entertainment Assist [34]	MHS	WT, O	N
Support Act Available online: https://supportact.org.au (accessed on 21 January 2024).	Hiruy [35], 2022	MHS	OU, MT, WT, IPSD	Y
Nursing and Midwifery Health Program Available online: https://www.nmhp.org.au (accessed on 10 January 2024).	Nursing and Midwifery Health Program Victoria [36], 2023	MHS	MT, IPSD	N
Minds Count Available online: https://mindscount.org (accessed on 22 January 2024).	Minds Count Foundation [37]	MHS	OU, O	N
Mentally healthy minimum standards for the creative, media and marketing industry Available online: https://www.mentally-healthy.org (accessed on 17 January 2024).	Mentally Healthy Change Group [38]	MHS	OU, O	N
Drs4Drs Available online: https://www.drs4drs.com.au (accessed on 19 January 2024).	Drs4Drs: An Australian Medical Association subsidiary [39]	MHS	IPSD, O	N
Arts Wellbeing Collective Available online: https://artswellbeingcollective.com.au (accessed on 20 January 2024).	The Arts Wellbeing Collective [40]	MHS	OU, MT, WT, IPSD, O	N
Healthy Heads in Trucks and Sheds Available online: https://www.healthyheads.org.au (accessed on 24 January 2024).	Healthy Heads in Trucks and Sheds [41] 2023	MHS	OU, MT, WT, IUPS, IUPA, IPSD, IPAD	Y
Ifarmwell Available online: https://ifarmwell.com.au (accessed on 24 January 2024).	Gunn, Skaczkowski [42] †, 2023	MHS	IUPS, IPSD	Y
Superfriend Available online: https://www.superfriend.com.au (accessed on 10 January 2024).	SuperFriend [43], 2023	MHS	OU, MT, WT, IUPS, O	Y
MATES in construction * Available online: https://www.mates.org.au/construction/ (accessed on 10 January 2024).	Ross et al. 2020 [44] † Gullestrup et al. 2023 [45] †, Maheen et al. 2022 [46] †	MHS	WT	Y
MATES in mining * Available online: https://www.mates.org.au/mining/ (accessed on 10 January 2024).	Sayers et al., 2019 [47] † Tynan et al., 2018 [48] †	MHS	WT	Y
MATES in energy * Available online: https://www.mates.org.au/energy/ (accessed on 10 January 2024).	Ross et al., 2020 [49] †	MHS	MT, WT	Y
MATES in manufacturing trial * Available online: https://www.amwu.org.au/mates_in_manufacturing (accessed on 15 January 2024).	Hall and Guntuku [50] 2023	MHS	WT	Y
Incolink Wellbeing & Support + BlueHats Available online: https://incolink.org.au/wellbeing-support-services (accessed on 19 January 2024).	King et al., [51] † 2023	Other	OU, WT, MT, IPSD	Y

^a^ PA = professional association; U = union; BA = business association; MHS = mental health-specific organisation; O = other. ^b^ OU = Organisational: universal. OMH = Organisational: workers with MH conditions. MT = manager training. WT = worker training. IUPS = worker training. IUPA = Individual Universal Physical Activity. IPSD = Individual: psychosocial programs for workers with emotional distress. IPAD = Individual Physical Activity for workers with emotional distress. O = other. * MATES in Construction is an industry-based suicide prevention program initiated in 2007. MATES has been extended to the mining, energy, and manufacturing industries in Australia. See [45] for an overview of peer-reviewed articles related to evaluations of MATES in construction, mining, and energy. † Indicates a peer-reviewed evaluation source.

## Data Availability

Not applicable.

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
