# Peer review of "What Is the Role of Industry-Based Intermediary Organisations in Supporting Workplace Mental Health in Australia? A Scoping Review"

_ijerph, 2025, doi:10.3390/ijerph22070974_

Round 1
Reviewer 1 Report
Comments and Suggestions for Authors
1-My main concern with this paper is its scientific basis. IJERPH is a scientific journal, and papers should provide solid scientific contributions in their respective fields. In the present study, of the final set of 35 ‘papers’ on which the scoping review is based, only 5 were peer-reviewed journal articles. Four items were published reports authored by external academic consultants, seven were published reports authored by internal program staff, and 19 were websites. I’m wondering about the solidity of the included items, particularly websites.
2-I understand that you applied guidelines for running scoping reviews. But this is not enough to make a relevant contribution. I could have also applied scoping review guidelines to newspaper articles, but this would not make a relevant contribution to a scientific journal.
3-Additionally, too many types of interventions are considered, and too many types of outcomes are also focused on (i.e., from mental health indicators to ROI), leading to a paper that does not have focus and does not help to understand what the role of intermediary organizations is as far as mental health interventions at work.
4-The utility of focusing on intermediary organizations is not clear. If the main focus is to understand the impact of interventions on mental health at work, why should we focus only on interventions provided by intermediary organizations? What is not clear is the link between delivering industry-specific rather than generic approaches to improving worker mental health, on the one hand, and intermediary organizations, on the other hand. I agree that the intervention should be specific, but I think that an intervention can be specific and well-grounded, independent of who carries out it. A stronger rationale should be provided here.
5-Why focus only on Australian intermediary organizations? This should be explained better.
Reviewer 2 Report
Comments and Suggestions for Authors
Please provide empirical evidence supporting the classification of workplace mental health interventions into two distinct approaches (p. 1, line 39-40), as referenced in the literature.
What factors informed the decision to focus specifically on industry intermediary organizations in Australia? Please elaborate on the distinctive characteristics of these organizations within the Australia context that make them suitable for this investigation.
What distinguishes scoping reviews from systematic reviews in methodology and purpose? What rationale supports the selection of a scoping review methodology for examining industry intermediary organizations? Additionally, what value does the inclusion of grey literature bring to the comprehensive understanding of industry intermediary organizations in this context?
Please include appropriate citations for the conceptual framework of industry intermediary organizations (p. 2, line 68-69), drawing from seminal works in this field.
Could you clarify what specific sectors or professional domains (p. 2, line. 70-71) are encompassed by the phrase 'particular industry or profession' in this context?
While the research objectives explicitly included exploring 'the opportunities that exist for industry intermediaries to support workers' mental health in the future (p. 2, line. 64)' this aspect appears underdeveloped in the current results section. Consider expanding this component to align with the stated research aims.
To strengthen comparative analysis, this research would benefit from examining interventions implemented by industry intermediary organizations in international contexts, which would provide valuable insights into transferable practices and context-specific considerations.
Reviewer 3 Report
Comments and Suggestions for Authors
Thank you for having me in this review. The manuscript presents a well-structured and insightful scoping review that addresses a relevant topic with clarity. However, before it can be considered for publication, the comments provided throughout this review should be thoroughly addressed to enhance the manuscript’s clarity, consistency, methodological transparency, and overall scholarly contribution.
Title: What is the role of industry-based intermediary organisations in supporting workplace mental health in Australia? A scoping review
Manuscript ID: ijerph-3632017
Abstract
The abstract is well-structured and coherent, clearly articulating the principal concepts of the study. However, it omits the reporting guideline (PRISMA-ScR) and lacks relevant keywords, which are essential for enhancing discoverability and aligning with standard scoping review practices.
Introduction
The introduction is well articulated, with the purpose of the study clearly identified and the research questions explicitly stated. However, it lacks a clear articulation of the study's contribution to the literature, which is essential for positioning the review within the broader academic discourse and demonstrating its novelty and relevance.
Materials and Methods
I expected the authors to justify why they used a scoping review over a systematic review, as such justification is essential to clarify the appropriateness of the chosen methodology in relation to the study’s objectives and scope.
The methodology is well positioned and articulated; however, the authors lose me at section 2.3 (Information Sources and Search Strategy), as I did not see the keywords used in the search process, an essential component for transparency and replicability.
I commend the authors for the explicit calculation and clear breakdown of the PRISMA flow diagram, which enhances the transparency and traceability of the study selection process in line with established reporting standards.
Results
The authors presented their findings well, supported by a clear table and informative figures, which enhance the readability and understanding of the mapped evidence and align with best practices for reporting scoping review results.
Discussion
The authors provided a clear discussion of their findings, effectively relating them to existing literature. However, from lines 340 to 391, they included implications, limitations, and directions for future research within the discussion section. While informative, these elements are typically more appropriately presented under distinct subheadings to improve clarity and structural coherence.
Conclusion
The conclusion should succinctly summarize the key findings, clearly highlight the study's contribution to the literature, and reinforce the relevance of the scoping review approach. It should also avoid introducing new information and ensure alignment with the objectives stated in the introduction.
Writing style
The authors predominantly used British English throughout the manuscript; however, at line 325, the word “Organizational” appears in American spelling. For consistency and professionalism, this should be revised to align with the chosen British writing style.

Round 2
Reviewer 1 Report
Comments and Suggestions for Authors
Thank you for considering my comments. I think the manuscript is now more clear.
I think an additional point to consider and mention is the lack of inclusion of tertiary interventions in the form of return-to-work programs. Why you didn't include these programs? What does it mean that they do not have a prevention component? They can prevent relapse, which is still prevention. Also, the WHO guidelines consider return-to-work programs (see recommendation 11 of the WHO report). Can you clarify a bit better? I'm also wondering whether this should be acknowledged as a further limitation of the study.
Thank you for considering my comments.
Author Response
Comment 1: I think an additional point to consider and mention is the lack of inclusion of tertiary interventions in the form of return-to-work programs. Why you didn't include these programs? What does it mean that they do not have a prevention component? They can prevent relapse, which is still prevention. Also, the WHO guidelines consider return-to-work programs (see recommendation 11 of the WHO report). Can you clarify a bit better? I'm also wondering whether this should be acknowledged as a further limitation of the study.
Response 1: We thank the reviewer for their careful consideration of this manuscript, and this constructive comment. As noted in the eligibility criteria section, programs focused solely on return-to-work were excluded from our review, however this was not acknowledged in the limitations section. In response to your suggestion, we have now expanded our rationale in the Limitations section to clarify that this decision aligns with our focus on upstream mental health strategies targeting primary and secondary prevention. We have also acknowledged that intermediaries may have a role in return-to-work initiatives and suggested this as an area for future research (see Lines 413-417).
Reviewer 2 Report
Comments and Suggestions for Authors
The author has thoroughly addressed the reviewer's comments, and the revised manuscript is now suitable for publication in IJERPH.
Author Response
Thank you
Reviewer 3 Report
Comments and Suggestions for Authors
Thank you for having me in this review for the second time. I have carefully examined the revised version, and I am pleased to note that the authors have adequately addressed all the concerns and comments raised during the initial review. The manuscript has significantly improved and is now well-structured, coherent, and ready for publication. I therefore recommend acceptance in its current form.

Author Response
Thank you